# Hearing Loss in Infants and Children with Asymptomatic Congenital Cytomegalovirus Infection: An Update in Diagnosis, Screening and Treatment

**DOI:** 10.3390/diagnostics15162026

**Published:** 2025-08-13

**Authors:** Yiyun Zhang, Yihan Ke, Mengwen Shi, Xiaoying Wang, Jie Yuan, Yu Sun

**Affiliations:** 1Department of Otorhinolaryngology, Union Hospital, Tongji Medical College, Huazhong University of Science and Technology, Wuhan 430022, China; 2Hubei Province Clinical Research Center for Deafness and Vertigo, Wuhan 430022, China

**Keywords:** congenital cytomegalovirus infection, sensorineural hearing loss, universal screening, antiviral therapy, delayed-onset hearing loss, valganciclovir, ganciclovir

## Abstract

Cytomegalovirus (CMV) represents the most prevalent cause of congenital viral infection in newborns and the leading non-genetic etiology of sensorineural hearing loss (SNHL) in children. Notably, only 10–15% of congenitally infected infants possibly present with classic clinical symptoms at birth, including Small for gestational age, Microcephaly, Petechiae or purpura, Blueberry muffin rash, Jaundice, Hepatomegaly, Splenomegaly and abnormal neurologic signs. In contrast, approximately 90% of infected neonates exhibit no apparent symptoms initially. Current research predominantly focuses on symptomatic cases due to their severe acute presentations and high rates of long-term sequelae (40–60%), including SNHL and neurodevelopmental impairments. However, significant controversy persists regarding the management of asymptomatic infants. Emerging evidence reveals that 8–15% of asymptomatic carriers develop Late-onset Hearing Loss (LOHL) beyond the neonatal period. Additionally, 5–10% may manifest neurodevelopmental abnormalities including mild intellectual disability, learning difficulties, or motor coordination disorders. Crucially, given the substantial population of asymptomatic cCMV cases, these delayed complications account for 30–40% of all cCMV-related long-term morbidity, underscoring their considerable public health impact. This review synthesizes current evidence and controversies regarding cCMV-related SNHL in asymptomatic or mildly symptomatic children, with a focus on screening, diagnostic classification, and antiviral management gaps, to heighten clinical awareness of this underrecognized cause of hearing loss.

## 1. Introduction

Cytomegalovirus (CMV), a member of the human herpesvirus family, represents the most prevalent viral etiology of congenital infections. The global prevalence of congenital CMV (cCMV) ranges between 0.2% and 2.5%, with higher seroprevalence observed among women, elderly populations, and populations in low-to-middle-income countries. CMV infection can impair hearing by inducing cochlear inflammation, damaging spiral ganglion neurons, disrupting Notch signaling, and altering inner ear development [1]. In MCMV-infected mice, reduced hair cell synapses, disorganized auditory fibers, and decreased neurite density indicate cochlear neuropathy [2]. Additionally, central nervous system symptoms in some patients suggest involvement of central auditory pathways. Cytomegalovirus (CMV) infection is associated with a relatively characteristic pattern of hearing loss, though significant individual variability exists. Sensorineural hearing loss (SNHL) is the most common form of auditory impairment in cCMV and may be unilateral or bilateral, with severity ranging from mild to profound, often exhibiting a fluctuating course [2]. cCMV infection is the leading preventable cause of childhood SNHL and a major non-genetic contributor to severe congenital malformations and neurodevelopmental delays [3]. cCMV infection manifests clinically detectable symptoms in 10–15% of neonates at birth, while 85–90% present without discernible clinical findings [4]. Although asymptomatic cases account for the majority of infections, they contribute to 30–40% of cCMV-related long-term morbidity and still lack standardized management approaches. The 2017 European consensus established diagnostic classifications: asymptomatic denotes absence of CMV-related signs (including Small for gestational age, Microcephaly, Petechiae or purpura, Blueberry muffin rash, Jaundice, Hepatomegaly, Splenomegaly, Microcephaly, Lethargy, Hypotonia, Seizures, or Impaired sucking reflex); mild encompasses isolated (≤2 transient/non-significant) manifestations; moderate involves persistent hematologic/biochemical abnormalities or ≥2 mild features; severe indicates central nervous system involvement or life-threatening conditions. Notably, classification of isolated SNHL remains contentious—most experts at the consensus meeting opposed antiviral treatment for mild isolated presentations due to insufficient RCT evidence supporting therapeutic benefit [5]. While symptomatic neonates with overt infection demonstrate clear treatment indications and higher risks of long-term sequelae, which have traditionally driven research efforts, recent investigations have increasingly shifted toward asymptomatic or mildly symptomatic cohorts. This shift acknowledges their substantial population burden: 10–15% develop delayed-onset SNHL [6], 5–10% exhibit neurodevelopmental impairments (intellectual disability, learning deficits, motor dysfunction) [7]. Emerging clinical perspectives advocate universal neonatal screening to capture this high-risk population and propose antiviral intervention for isolated SNHL to mitigate sequelae, though these approaches remain debated. This review synthesizes current perspectives on diagnosis, screening, treatment, and follow-up of hearing loss in children with asymptomatic or mild cCMV infection, highlighting controversies, critical knowledge gaps, and actionable research priorities to inform management strategies.

## 2. Epidemiology

Cytomegalovirus (CMV) demonstrates a global birth prevalence of 0.5–1.3% [1]. Seroprevalence reaches 58–79% among North American women of childbearing age and ~86% worldwide [1,8]. Higher seropositivity is observed in females, older populations, and low-to-middle-income countries. The infection affects 1/200 live births in high-income nations compared to 1/71 in resource-limited settings [8]. CMV transmission occurs via vertical, horizontal, and sexual routes [9,10]. Postnatally acquired infections typically result from contact with infected bodily fluids (urine, saliva, semen, cervical secretions, breast milk) or medical procedures solid organ/allogeneic hematopoietic cell transplantation (HCT), transfusion-mediated transmission is relatively uncommon [11]. cCMV infection primarily stems from maternal vertical transmission during either primary infection (24–40% fetal transmission risk) or non-primary infection (0.5–2% transmission risk) [12,13,14]. Definitive diagnosis of cCMV requires viral detection within 21 postnatal days [15], as later identification may indicate postnatal acquisition, frequently via breast milk. SNHL develops in 40–60% of symptomatic and 10–14% of asymptomatic cCMV-infected infants [16], manifesting unilaterally or bilaterally [17]. Notably, 10–20% of cCMV-associated SNHL cases present with late-onset, exhibiting threshold fluctuations or progression [18]. While First-trimester maternal infection confers substantially elevated risks of severe SNHL, third-trimester exposures may induce mild auditory deficits. Importantly, the potential for severe neurological sequelae remains exclusively associated with maternal infections occurring during the initial gestational trimester [1,19,20].

## 3. Diagnosis and Screening

cCMV infection is implicated in approximately 10% of childhood cerebral palsy cases [21] and 25% of SNHL cases [22]. However, only 15% of infected infants exhibit clinical manifestations at birth [11,19]. The majority of cases remain undetected unless severe symptoms emerge, resulting in fewer than 25% identification within the first postnatal month [5]. Among those classified as asymptomatic at birth, up to 20% develop late-onset SNHL, developmental delays, or neurocognitive impairments, with diagnoses typically delayed beyond 12 months of age [23,24,25]. Early identification is critical, as delayed diagnosis may exacerbate adverse neurodevelopmental outcomes.

Three neonatal screening strategies exist for cCMV: targeted screening, expanded targeted screening, and universal screening. Testing within 21 days of birth is essential to differentiate congenital from postnatal infections, which pose lower risks of SNHL and developmental deficits [5,19]. The 2017 European consensus recommends CMV testing only for symptomatic neonates or those with confirmed SNHL, primarily due to cost constraints, management uncertainties for asymptomatic cases, and parental anxiety concerns [5,26]. This approach misses >75% of asymptomatic infants. Universal screening has been advocated as the sole method capable of enabling early intervention for all cCMV-positive infants at risk of late-onset hearing loss [27]. Notably, the American Academy of Otolaryngology–Head and Neck Surgery (AAO-HNS) issued a 2024 position statement endorsing universal cCMV newborn screening, emphasizing its potential to improve hearing outcomes through early diagnosis (https://www.entnet.org/resource/universal-newborn-congenital-cytomegalovirus-ccmv-screening/ (accessed on 10 May 2025)). Recent large-scale studies provide supporting evidence: Leruez-Ville et al. [28] identified a 0.37% cCMV prevalence (51 confirmed cases) among 11,740 screened neonates, with 4 infants developing unilateral/bilateral SNHL. Similarly, Barkai et al. [29] reported a 0.49% prevalence (47 confirmed cases) in 9845 neonates, where initial pass rates on newborn hearing screening failed to predict subsequent SNHL detected during follow-up. These findings underscore universal screening’s capacity to identify at-risk asymptomatic infants requiring longitudinal audiological monitoring.

Dried blood spot (DBS) PCR has emerged as a practical tool for universal screening, leveraging existing newborn metabolic screening infrastructure [30]. DBS offers logistical advantages including cost-effectiveness, ease of storage, and retrospective diagnostic potential for late-onset SNHL. Minnesota became the first U.S. state to implement universal DBS-based cCMV screening in 2023, with 13 additional states (including New York) subsequently adopting similar programs, though methodological standardization remains lacking [31,32]. However, DBS PCR has lower sensitivity than saliva PCR (75% vs. >98%), potentially missing up to one-third of cases due to lower viral DNA loads in blood [30,33]. Boppana et al. [30] prospectively compared single- versus dual-primer DBS PCR with rapid saliva culture, revealing that even optimized dual-primer protocols failed to detect 80% of cCMV cases. While recent DBS sensitivity improvements (75% detection rate) represent progress, one-quarter of infections remain undiagnosed [31], necessitating methodological refinements. Conversely, saliva PCR achieves >98% sensitivity but faces challenges with false positives from breastmilk CMV DNA contamination, requiring confirmatory urine PCR [19,21,34]. Cost barriers further limit saliva PCR’s scalability for universal screening [35]. Israel’s pooled saliva testing strategy addresses these limitations by batch-testing multiple samples followed by individual retesting of positive pools, achieving a 55.6% detection rate for asymptomatic cCMV cases missed by targeted screening [36].

For the diagnosis of cCMV, there is currently no clearly unified gold standard. However, The ESPID consensus guidelines state that cCMV should be diagnosed in infants within 21 days of birth via urine or saliva PCR, recommending a preference for urine over saliva for CMV PCR testing [5]. This is because saliva PCR can yield false positives due to CMV shedding in breast milk. If neonatal urine collection is difficult, saliva PCR may be used, but a positive saliva PCR result should be confirmed with urine PCR [5]. In clinical practice, the necessity of combining DBS PCR with saliva/urine PCR for universal screening lies in: 1. Enhanced Sensitivity and Specificity. While DBS PCR leverages existing newborn screening infrastructure in many countries, requiring only additional PCR analysis without extra sampling, and offers advantages like lower cost, ease of storage and transport for mass screening, its sensitivity remains suboptimal. Combining it with saliva/urine PCR helps mitigate the risks of missed diagnoses (false negatives) or misdiagnoses (false positives) inherent in relying on a single method. 2. Broader Population Coverage. CMV shedding in urine and saliva of children with cCMV may not persist indefinitely, making these tests less reliable in older children (beyond 21 days of birth) [37]. For this population, CMV PCR can be performed on residual dried blood spots stored from newborn screening. Without available DBS, a definitive diagnosis of cCMV cannot be made for these older children [31].

Cost-effectiveness analyses yield conflicting perspectives. A study conducted in Singapore compared the effectiveness of three screening strategies over a two-year period and estimated the total direct costs to be SGD 104,445.79 (approximately USD 77,289.88) for no screening, SGD 146,656.30 (approximately USD 108,527.66) for targeted screening, and SGD 853,890.16 (approximately USD 631,878.72) for universal screening. Similarly, Gantt et al. [38] calculated per-case identification costs of USD 566–2832 (targeted) versus USD 2000–10,000 (universal). However, when accounting for lifetime costs of undiagnosed SNHL, universal screening demonstrated greater net savings in most models [38,39]. A Japanese decision-tree analysis further supported universal screening’s superior cost-effectiveness compared to targeted approaches [39].

Prenatal diagnosis requires early first-trimester maternal serology testing (IgG/IgM + IgG avidity), as fetal sequelae correlate strongly with first-trimester transmission [40]. The cytomegalovirus (CMV) IgG avidity index (AI) serves as an important indicator of the timing of infection. A low AI suggests a recent primary infection, whereas a high AI is more indicative of a past or recurrent infection. In general, an AI greater than 60% is consistent with an infection that occurred more than three months prior, reflecting either a previous or reactivated infection. Conversely, an AI below 30% strongly suggests a primary infection within the past three months. When the AI falls between 30% and 60%, interpretation requires a comprehensive assessment that includes CMV IgM and IgG serologies as well as the patient’s clinical history [41]. CMV IgG/IgM serology with IgG avidity testing confirms primary infection in IgM-positive cases. Amniotic fluid PCR after 17 weeks is the gold standard [21]. Late-gestation fetal MRI and ultrasound provide prognostic insights, demonstrating high negative predictive values (96.9–98.5%) for severe outcomes when normal. Notably, 46% of asymptomatic neonates exhibit abnormal neuroimaging, though imaging poorly predicts late-onset SNHL [23].

For children with congenital cytomegalovirus (cCMV) infection, initial hearing screening should be conducted within the first month after birth. This includes auditory brainstem response (ABR) testing, which assesses the function of the auditory neural pathways; distortion product otoacoustic emissions (DPOAE), which evaluate the function of the outer hair cells in the cochlea; and tympanometry, which examines middle ear ventilation function [42]. If the initial screening is passed, regular follow-up using otoacoustic emissions (OAE) should be conducted at 5 months, 1 year, 2 years, 3 years, and 4 years of age. If the initial screening is not passed, further evaluation should include ABR testing (to objectively determine hearing thresholds) and behavioral audiometry, where applicable, for children who are able to cooperate [42]. Long-term follow-up is recommended through at least 6 years of age, using ABR in combination with behavioral audiometry (BA). After age 6, the testing approach should be adjusted based on the child’s rehabilitative needs [43,44].

For delayed-onset SNHL beyond 21 days, retrospective DBS analysis confirms cCMV. Differential diagnosis incorporates maternal infection history, characteristic neuroimaging findings (e.g., periventricular calcifications), and exclusion of genetic, anatomical, or alternative infectious etiologies to optimize diagnostic accuracy. The detailed diagnosis and treatment process, summarized based on a comprehensive review of current literature, is shown in Figure 1.

However, despite the numerous advantages of universal cCMV screening, there are still some challenges in the subsequent hearing screening process. One key issue is the significant variation in the onset time of SNHL, which requires multiple hearing assessments over time, placing a certain burden on both families and the healthcare system. Parents often lack an adequate understanding of the importance of follow-up appointments and may miss scheduled visits, particularly among populations facing socioeconomic barriers or limited access to healthcare resources. In busy hospital settings, managing the time required for follow-up appointments for patients is also a practical challenge. These obstacles highlight the need to consider real-world constraints when implementing universal screening programs.

## 4. Classification

The 2017 European Consensus Committee defines isolated SNHL as asymptomatic cCMV infection in the absence of concurrent clinical, laboratory, or neuroimaging abnormalities [19]. In contrast, the European Society for Paediatric Infectious Diseases (ESPID) classifies isolated CMV-associated SNHL as a manifestation of CNS involvement [5].

Table 1 compares evolving classification frameworks for cCMV-associated outcomes, highlighting discrepancies between the 2017 ESPID criteria, the 2024 European Consensus on Clinical Investigation (ECCI), and recent perspectives [5,12,21].

## 5. Prevention and Treatment

### 5.1. Three-Tiered Prevention Measures for cCMV Infection

Primary prevention spans the entire pregnancy period, particularly preconception and early gestation, including vaccination and behavioral prophylaxis. Secondary prevention targets primary CMV infection during pregnancy, especially in the first trimester, involving antivirals, hyperimmune globulin, and monoclonal antibodies. Tertiary prevention involves using valacyclovir during pregnancy after amniocentesis-confirmed fetal CMV infection to reduce maternal viral load and vertical transmission risk. Confirmed fetal CMV infection may present as: asymptomatic, mild/isolated symptoms, or moderate/severe clinical symptoms at birth (refer to Table 1 for details).

### 5.2. Antiviral Treatment for Moderate/Severe Symptoms

Since 2017, antiviral therapy has been recommended for infants with moderate/severe symptoms [5,19,21]. Kimberlin et al. compared 6-week intravenous ganciclovir versus no treatment for hearing outcomes in neonates [45]. Among 100 enrolled infants, 47 were randomized to the treatment group and 50 to the control group. The treatment group received ganciclovir (6 mg/kg IV q12h for 6 weeks), while controls received no treatment. At 6 months: 84% (21/25) treated vs. 59% (10/17) controls improved/maintained hearing; 0% vs. 41% deteriorated. At ≥1-year follow-up, 21% (5/24) of treated versus 68% (13/19) control patients showed hearing deterioration. Grade 3/4 neutropenia: 63% (29/46) treated vs. 21% (9/43) controls (*p* < 0.01).

Yang et al. [46] conducted an RCT comparing 6-week IV ganciclovir (control) versus oral valganciclovir (intervention). Both groups showed significant reductions in abnormal hearing rates (20.83% control vs. 29.17% intervention). Kimberlin et al. [47] conducted a pharmacokinetic comparison between intravenous ganciclovir and oral valganciclovir, demonstrating that the oral valganciclovir had lower neutropenia risk than IV ganciclovir. A subsequent study compared 6-week vs. 6-month oral valganciclovir [48]. At the 6-month assessment, both treatment groups exhibited comparable improvements in best-ear hearing outcomes. By 12 months, however, the 6-month treatment group demonstrated a higher proportion of ears maintaining or achieving normal hearing across all auditory frequencies compared to the 6-week cohort. Longitudinal follow-up at 24 months further revealed superior neurodevelopmental scale scores in the 6-month treatment group, underscoring the sustained benefits of prolonged antiviral therapy. While both regimens showed comparable short-term hearing improvement, the longer course appeared to moderately improve long-term hearing and developmental outcomes [48]. Current clinical practice typically employs 6-month oral valganciclovir for symptomatic cCMV, with individualized adjustments. Overall, for infants with moderate/severe cCMV, antiviral therapy (IV ganciclovir or oral valganciclovir) can improve/maintain hearing and reduce deterioration risk, despite neutropenia side effects. However, ganciclovir/valganciclovir use remains controversial for asymptomatic/mild cases [49].

### 5.3. Management of Asymptomatic/Mild Cases

In the 2017 European consensus, most experts believe that asymptomatic children and children with mild isolated symptoms should not receive treatment, and there is no consensus on whether to treat isolated SNHL [5]. The risk-benefit ratio of antiviral therapy must be carefully considered. Some cohort studies support potential benefits. Chung et al. conducted a non-randomized trial comparing valganciclovir versus no treatment in minimally symptomatic infants [50]. Among 37 analyzed participants (25 treatment, 12 control), controls were more likely to experience hearing decline. Mean best-ear hearing decreased by 13.7 dB in controls versus improved by 3.3 dB in the treatment group. Neutropenia occurred in 3/20 treated versus 1/8 control infants.

Lanzieri et al. [51] prospectively followed 92 asymptomatic cCMV infants and 51 controls from 3 months to 5 years. SNHL prevalence increased from 7% to 14% in cases, versus 0% in controls; from 6–18 years, changes were 11% versus 8%, respectively. By age 18, SNHL prevalence reached 25% (95% CI: 17–36%) in cases versus 8% (95% CI: 3–22%) in controls. Cases with unilateral congenital/early-onset SNHL had higher risk of delayed-onset loss in the initially normal ear. Among affected ears, 65% showed progressive deterioration, including 40% of better ears in bilateral cases.

Pasternak et al. [52] retrospectively studied 59 infants with isolated SNHL receiving valganciclovir/ganciclovir. Of 80 affected ears at baseline, 68.8% (55) improved (96.3% normalized), 2.5% (2) worsened; among 21 infants with bilateral loss, 76.1% (16) showed improvement (93.7% achieving normal functional hearing). Neutropenia (ANC) is typically characterized by a neutrophil count in the blood below 1.5 × 10^9^/L (1500/mm^3^) [53]. ANC ≤1000/mm^3^ occurred in 19 infants (30 episodes), primarily during the first 3 months, including one grade 3 case (ANC 400–599 cells/mm^3^) [52].

Lackner et al. [54] followed 18 treated asymptomatic children; 89% (16) maintained normal hearing at 4–10 years. Turriziani Colonna et al. [43] reported no SNHL in 24 treated asymptomatic infants followed to 41.5 months. Villaverde et al. [55] retrospectively analyzed 196 mild cCMV infants (68 treated). Overall SNHL incidence at 24 months was 4.6% (9/196), with no difference between treated (4.4%) and untreated (4.7%) groups (*p* = 0.658). Long-term drug effects and post-24-month hearing outcomes remain unclear. Çiftdoğan et al. [49] reported a case of asymptomatic cCMV with bilateral SNHL showing CMV pp65 antigenemia reduction after 1-week ganciclovir, undetectable levels after 5-week valganciclovir, and progressive SNHL improvement over 1-year follow-up. These observational data demonstrate hearing protection despite significant neutropenia risks (63% ganciclovir vs. 12% valganciclovir).

### 5.4. Delayed-Onset Hearing Loss

Amir et al. [56] analyzed 21 infants with delayed-onset SNHL receiving ganciclovir/valganciclovir. Among 42 affected ears pretreatment: 52% (22) mild, 24% (10) moderate, 7% (3) severe loss. Post-treatment, only 5% (2) remained in each category, with 86% (36) normalizing and 69% (29) improving. Stronati et al. [57] reported an infant with normal newborn hearing who developed isolated left mild SNHL at 3 months. Without treatment, BERA at 6 months showed moderate bilateral deficit. After 6-week valganciclovir (15 mg/kg q12h), hearing normalized by 12 months. However, given the fluctuating nature of cCMV-related hearing loss, spontaneous recovery must be distinguished from treatment effects through long-term follow-up.

### 5.5. Current Perspectives

Current evidence suggests potential therapeutic benefits of antiviral treatment for children with isolated or delayed-onset SNHL. Although valganciclovir’s short-term toxicities (neutropenia, hepatotoxicity) are usually reversible, they require frequent hematologic monitoring [58]. The sustained benefits of neonatal antiviral therapy beyond age 2 remain unclear [59], as limited longitudinal follow-up data and insufficient safety surveillance preclude robust conclusions regarding universal intervention for isolated/delayed SNHL. Critical knowledge gaps persist, including: (1) the identification of reliable biomarkers for delayed-onset SNHL progression [60]; (2) comprehensive pharmacokinetic, efficacy, and safety profiling of oral valganciclovir regimens [49]; and (3) evidence-based criteria for initiating treatment in asymptomatic or minimally symptomatic cases. These unresolved issues underscore the imperative for targeted clinical investigations to optimize therapeutic protocols.

## 6. Follow-Up

Therapeutic management of cCMV infection has historically been constrained by limited long-term follow-up data. Prior studies lacked prospective audiologic evaluations beyond age 2 in infants with symptomatic cCMV disease receiving six-month valganciclovir (VGCV) therapy [44]. Recently, a Japanese investigator-initiated, single-arm, prospective multicenter trial evaluated long-term hearing outcomes in symptomatic cCMV infants treated with oral VGCV (16 mg/kg bid for six months), with three-year follow-up. At six months, 19 of 48 ears (40.0%) demonstrated hearing improvement, increasing to 27 ears (56.3%, *p* = 0.032) by three years. The proportion of ears achieving and sustaining normal hearing rose from 25 (52.5%) at six months to 35 (72.9%, *p* = 0.011) at three years, while stable hearing preservation rates remained comparable between timepoints. No delayed adverse effects were observed post-treatment [44]. These findings by Morioka et al. suggest a favorable benefit-risk profile for antiviral therapy in maintaining or improving auditory function in symptomatic cases.

Delayed-onset hearing loss (DHL) is defined as postnatally acquired hearing deficits in infants passing newborn hearing screening programs (NHSP) [61]. Progressive hearing loss—deterioration to a more severe hearing category in ≥1 ear—also falls under DHL. Notably, 8–12% of asymptomatic cCMV infants develop isolated SNHL [62]. Approximately 50% of children diagnosed with hearing loss by the age of 10 had passed the Newborn Hearing Screening Program (NHSP), and undiagnosed and untreated delayed-onset hearing loss (DHL) can lead to speech and comprehension delays, adversely affecting a child’s development, language acquisition, communication skills, and cognitive abilities, with serious negative impacts on overall growth [61]. While current prenatal or neonatal tools cannot reliably predict DHL, neuroimaging markers of central nervous system involvement (e.g., ventriculomegaly, white matter lesions, or calcifications) correlate with elevated SNHL risk. Most existing studies feature ≤2-year follow-up periods, potentially missing late-onset DHL, while infrequent audiologic assessments may delay detection of early hearing declines [43,63].

According to longitudinal data by Lancrer [54], Fourgeaud [7], and Liesbeth [64], 40–60% of symptomatic infants exhibit SNHL at birth (71% severe-to-profound), with 37.5% experiencing further auditory deterioration during follow-up, often necessitating cochlear implantation. Among asymptomatic cases, 8–15% develop SNHL between 6 months and 6 years of age (50% unilateral; fluctuating/progressive in subset), yet 43% discontinue follow-up before age 3, increasing DHL underdiagnosis [7,54,64]. Long-term audiologic follow-up is recommended for all children with cCMV infection. In addition, infants with specific risk factors—such as a history of neonatal intensive care unit (NICU) stay exceeding five days—should undergo hearing evaluations beginning at 3 months of age and continuing annually until at least 3–4 years of age [61]. Early detection and timely intervention are critical in mitigating the impact of DHL. For asymptomatic infants who develop or show progression of DHL, selective oral VGCV therapy may be considered, accompanied by close toxicity monitoring. However, the optimization of preventive and therapeutic strategies in this population remains dependent on robust long-term follow-up data.

## 7. Conclusions

The management of cCMV infection associated hearing loss necessitates a dual approach integrating early detection and evidence-based interventions. Current evidence supports antiviral therapy—specifically a six-month course of oral valganciclovir—for improving auditory and neurodevelopmental outcomes in symptomatic infants [15,65]. However, its use in asymptomatic or minimally symptomatic cases remains controversial. While observational studies suggest potential benefits in attenuating hearing deterioration [66], the absence of high-quality randomized controlled trial (RCT) data, coupled with risks of hepatotoxicity and other adverse effects, limits therapeutic consensus for this population [11,19]. Universal screening is currently the most effective measure to reduce the rate of missed diagnosis of asymptomatic children. Although some developed countries have begun to implement universal screening, its implementation still needs to make a balance between low screening cost and high positive rate, as well as the development of standardized management. The high rate of missed diagnosis of delayed-onset hearing loss exposes the shortcomings of the current follow-up system, and there is a need to formulate follow-up strategies and develop more methods to predict the risk of long-term sequelae.

## Figures and Tables

**Figure 1 diagnostics-15-02026-f001:**
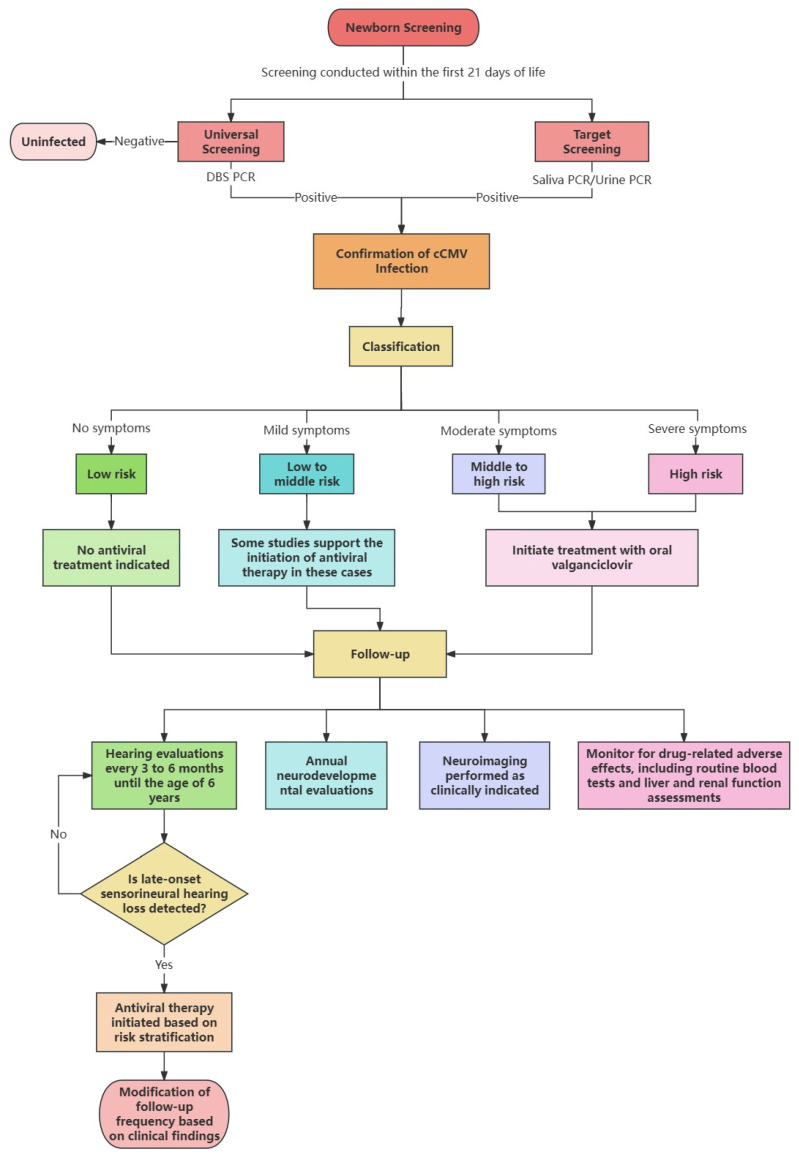
Flowchart for screening, diagnosis and treatment of congenital cytomegalovirus infection.

**Table 1 diagnostics-15-02026-t001:** The difference between the 2017 ESPID criteria and recent perspectives.

Classification Criteria	2017 ESPID (Luck et al.)	2024 ECCI (Leruez-Ville et al.) and Recent Views	Updates/Controversies
Severe Disease Definition	CNS involvement (microcephaly, neurologic signs, calcifications) Life-threatening organ failure Bilateral SNHL	**High-risk cCMV:**	Added “maternal primary infection in 1st trimester” as independent risk. isolated SNHL reclassified to moderate-risk.
		Maternal primary infection in 1st trimester Severe CNS lesions (ventriculomegaly, cortical malformations on MRI) Multisystem involvement	
Moderate Disease Definition	Persistent laboratory abnormalities (>2 weeks) ≥2 mild clinical features (e.g., jaundice + hepatosplenomegaly)	**Moderate-risk cCMV:**	Isolated SNHL now categorized as moderate-risk (previously “severe”). lab abnormalities require viral load correlation.
		Isolated SNHL (unilateral/bilateral) Non-CNS persistent abnormalities (hepatitis, thrombocytopenia)	
Mild Disease Definition	Isolated/transient findings (petechiae, mild hepatomegaly)	**Low-risk cCMV:**	“Mild disease” category removed. asymptomatic infants with normal imaging classified as low-risk (no treatment).
		Maternal non-primary or 3rd trimester infection Asymptomatic with normal neuroimaging	
Asymptomatic Infection Definition	No clinical/laboratory abnormalities	**Expanded to:**	Controversy: Whether to treat subclinical imaging abnormalities (e.g., white matter changes) remains debated.
		Asymptomatic + normal imaging → low-risk Asymptomatic + minor imaging anomalies → moderate-risk (requires monitoring)	
Treatment Recommendations	Severe/moderate disease: 6-month antiviral therapy Mild/asymptomatic: No treatment	High-risk: 6-month valganciclovir Moderate-risk (SNHL): 6-month therapy Moderate-risk (non-CNS): 6-week therapy Low-risk: No treatment	Controversy: Prophylactic treatment for asymptomatic infants with imaging anomalies lacks RCT evidence.
Role of Neuroimaging	CrUSS (cranial ultrasound) as initial screening; MRI only if abnormalities detected	**Mandatory MRI:**	Debate: MRI accessibility vs. cost. new neuroimaging scoring systems
		All high/moderate-risk cases Standardized scoring systems (e.g., white matter grading)	
Maternal Infection Impact	No distinction between primary/non-primary maternal infections	**Independent risk factor:**	Clear stratification by maternal infection type. asymptomatic infants with 1st trimester exposure require long-term follow-up.
		Maternal primary infection (especially 1st trimester) → high-risk Non-primary → low or moderate-risk	

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
