# Peer review of "Hearing Loss in Infants and Children with Asymptomatic Congenital Cytomegalovirus Infection: An Update in Diagnosis, Screening and Treatment"

_diagnostics, 2025, doi:10.3390/diagnostics15162026_

Round 1

Reviewer 1 Report

Comments and Suggestions for Authors

This manuscript reviews the issue of hearing loss in infants and children with asymptomatic congenital cytomegalovirus (CMV) infection. The topic of this review is interesting and important because it reminds us of the hearing loss associated with CMV infection.

However, there are several significant criticisms that must be addressed.

  1. Please clarify the necessity of combining universal DBS PCR with saliva/urine PCR for newborn CMV infection screening. Has urine PCR already become the gold standard?
  2. Regarding the IgG avidity index, the threshold should be specified as either <30% or 60%. Should the text mention the threshold of 30% or 60%?
  3. Is there a specific pattern of hearing loss following CMV infection? Does the infection affect the peripheral cochlear organ, the cochlear nerve, or the central auditory pathways and cortex? Address this information. 

Author Response

Response to Reviewer 1 Comments

1. Summary

We sincerely appreciate the positive feedback and insightful comments provided by the reviewer. We are pleased to hear that the reviewer found our article valuable in elucidating recent advances in the diagnosis, screening, and treatment of hearing loss in infants and children with asymptomatic congenital cytomegalovirus infection. We also appreciate the reviewer’s observations on areas for improvement, and we are fully committed to addressing these concerns to enhance the quality of our review. In response, we have carefully considered the suggestions and made the necessary revisions to ensure that the article becomes a more robust contribution to the field. Please find our detailed responses below, along with the corresponding revisions highlighted in red in the resubmitted files for your convenience.

2. Point-by-point response to Comments and Suggestions for Authors

Comments 1: Please clarify the necessity of combining universal DBS PCR with saliva/urine PCR for newborn CMV infection screening. Has urine PCR already become the gold standard?

Response 1: We sincerely thank the reviewer for this valuable suggestion. In response, we have revised Section 3 (Diagnosis and Screening) to include an additional paragraph highlighting the importance of combining universal DBS PCR with saliva/urine PCR in cCMV screening. We also explicitly emphasize that, although there is currently no universally accepted gold standard for CMV diagnosis, urine PCR remains the most sensitive method for detecting CMV infection in infants within the first 21 days after birth.

Comments 2: Regarding the IgG avidity index, the threshold should be specified as either <30% or 60%. Should the text mention the threshold of 30% or 60%?

Response 2: We thank the reviewer for this excellent suggestion. In response, we have revised Section 3 to incorporate the threshold values of the CMV IgG avidity index (AI) in diagnosing the type of maternal infection, as suggested. Specifically, we have elaborated on the following key points: The cytomegalovirus (CMV) IgG avidity index (AI) serves as a useful marker in distinguishing recent primary infection from past or recurrent infection. A lower AI indicates a more recent primary infection, while a higher AI suggests a prior or reactivated infection. In general, an AI greater than 60% is indicative of an infection that occurred more than three months earlier, consistent with past or non-primary infection. Conversely, an AI below 30% strongly points to a primary infection within the past three months. When the AI falls between 30% and 60%, interpretation should be guided by a combination of CMV IgM and IgG serologies, along with a detailed clinical history .

Comments 3: Is there a specific pattern of hearing loss following CMV infection? Does the infection affect the peripheral cochlear organ, the cochlear nerve, or the central auditory pathways and cortex? Address this information.

Response 3: We sincerely thank the reviewer for this insightful suggestion. In response, we have expanded the Introduction section to include a description of the potential effects of CMV infection on peripheral cochlear structures, the cochlear nerve, and the central auditory pathways and cortex, as well as the underlying mechanisms involved. We also emphasized that cytomegalovirus (CMV) infection is associated with a relatively characteristic pattern of hearing loss, although considerable inter-individual variability exists. Sensorineural hearing loss (SNHL) is the most common type of auditory impairment caused by congenital CMV (cCMV) infection. It may affect one or both ears, ranging from mild to profound in severity, and often presents with a fluctuating course. Some affected neonates exhibit hearing loss at birth, while others develop delayed-onset hearing loss months or even years later.

Reviewer 2 Report

Comments and Suggestions for Authors

Dear authors,

Thank you for submitting your manuscript about CMV.

Here are my recommendations:

60: is ref 25 only the source of "... develop delayed-onset SNHL" or for all data presented in this sentence? If for all please move the [25] after "burden" or at the sentence's end. If no, please add more refs.

71: Is [13] also the source? If yes please add also in the first sentence.

106ff: please explain in more detail the problem with hearing screening (not only in the US) due to the later onset including a large variation leading to multiple re-screening appointments. Problems of attendance and problem of time slots e.g. in the hospitals.

138ff: Please also give a converted value for the S$. I have no idea how much it is.

Fig. 1: It should be very clear whether this is a recommendation given by this manuscript or my one or several references. Please clarify.

205f: once introducted cCMV should be used always instead of "congenital CMV".

229: define ANC.

237: Please remove pre-names here and elsewhere in simlar citations.

243ff: Please expand this important part about HL. Also discuss which test method(s) is / are the best for screening in these children like AABR, ABR, ASSR, OAE, tympanogram, ... and in which combination and in which scenarios.

280: the DHL is the main problem at ENT departments and needs to be addressed in more detail.

Author Response

1. Summary

We sincerely appreciate the positive feedback and insightful comments provided by the reviewer. We are pleased to hear that the reviewer found our article valuable in elucidating recent advances in the diagnosis, screening, and treatment of hearing loss in infants and children with asymptomatic congenital cytomegalovirus infection. We also appreciate the reviewer’s observations on areas for improvement, and we are fully committed to addressing these concerns to enhance the quality of our review. In response, we have carefully considered the suggestions and made the necessary revisions to ensure that the article becomes a more robust contribution to the field. Please find our detailed responses below, along with the corresponding revisions highlighted in red in the resubmitted files for your convenience.

2. Point-by-point response to Comments and Suggestions for Authors

Comments 1: 60: is ref 25 only the source of "... develop delayed-onset SNHL" or for all data presented in this sentence? If for all please move the [25] after "burden" or at the sentence's end. If no, please add more refs.

Response 1: Thank you for your question. Reference [25] supports only the part “…develop delayed-onset sensorineural hearing loss (SNHL).” Additional reference [46] has been included to support the rest of the data in the sentence. We have also revised the sentence by removing “And 30–40% ultimately experience chronic complications” to ensure data accuracy and scientific rigor.

Comments 2: 71: Is [13] also the source? If yes please add also in the first sentence.

Response 2: Thank you for your question. Reference [13] does not support the first sentence. In response, we have added reference [14] to support the data presented in the first sentence of line 71.

Comments 3: 106ff: please explain in more detail the problem with hearing screening (not only in the US) due to the later onset including a large variation leading to multiple re-screening appointments. Problems of attendance and problem of time slots e.g. in the hospitals.

Response 3: Thank you for your valuable suggestion. We appreciate your recommendation to elaborate on the limitations of hearing screening in the context of cCMV. In the revised manuscript, we have expanded the discussion on the challenges associated with hearing follow-up, including the high variability in the onset of sensorineural hearing loss (SNHL), which often necessitates repeated audiological assessments over an extended period. We have also discussed issues related to low follow-up attendance rates, which may be influenced by socioeconomic barriers, parental awareness, and systemic constraints within healthcare settings. These additions aim to provide a more comprehensive view of the global challenges surrounding hearing screening in newborns with cCMV.

Comments 4: 138ff: Please also give a converted value for the S$. I have no idea how much it is.

Response 4: Thank you for your question. The approximate exchange rate used is 1 SGD ≈ 0.74 USD. Currency conversions have been added to the text as follows: No screening: S$104,445.79 (≈ $77,289.88), Targeted screening: S$146,656.30 (≈ $108,527.66),Universal screening: S$853,890.16 (≈ $631,878.72).

Comments 5: Fig. 1: It should be very clear whether this is a recommendation given by this manuscript or my one or several references. Please clarify.

Response 5: Thank you for pointing this out. We agree that clarification is necessary. Figure 1 is not an original clinical guideline but rather a summary of diagnostic and treatment approaches synthesized from the literature reviewed in this manuscript. The manuscript has been revised accordingly to clearly state this.

Comments 6: 205f: once introducted cCMV should be used always instead of "congenital CMV".

Response 6: Thank you for your detailed feedback. We fully agree, and have ensured that “cCMV” is used consistently throughout the manuscript after its first appearance (where the full term is defined).

Comments 7: 229: define ANC.

Response 7: Thank you for your suggestion. We have provided a clear definition of ANC in the text and added reference [83] for further clarification.

Comments 8: 237: Please remove pre-names here and elsewhere in simlar citations.

Response 8: Thank you for pointing out this lack of clarity. We have revised the manuscript by removing the given names from author citations throughout and retaining only their surnames.

Comments 9: 243ff: Please expand this important part about HL. Also discuss which test method(s) is / are the best for screening in these children like AABR, ABR, ASSR, OAE, tympanogram, ... and in which combination and in which scenarios.

Response 9: We thank the reviewer for this insightful suggestion. We have expanded this important section on hearing loss (HL) and clearly outlined appropriate testing methods for initial screening, diagnosis of HL/non-HL, and long-term monitoring in children with cCMV. This expanded content has been incorporated into Section 3 (Diagnosis and Screening).

Comments 10: 280: the DHL is the main problem at ENT departments and needs to be addressed in more detail.

Response 10: We thank the reviewer for pointing out this ambiguity. We have provided a more detailed discussion of DHL in the Follow-up section. We emphasized its significance within otolaryngology and highlighted the urgent need for ongoing management and monitoring strategies to address this issue effectively.